# Added Value of Biophysics to Study Lipid-Driven Biological Processes: The Case of Surfactins, a Class of Natural Amphiphile Molecules

**DOI:** 10.3390/ijms232213831

**Published:** 2022-11-10

**Authors:** Guillaume Gilliard, Aurélien L. Furlan, Willy Smeralda, Jelena Pršić, Magali Deleu

**Affiliations:** Laboratoire de Biophysique Moléculaire aux Interfaces, Gembloux Agro-Bio Tech, Université de Liège, 2 Passage des Déportés, B-5030 Gembloux, Belgium

**Keywords:** surfactin, biomimetic membrane models, molecular biophysics, lipid-mediated sensing, membrane interactions, biological mechanism

## Abstract

The role of membrane lipids is increasingly claimed to explain biological activities of natural amphiphile molecules. To decipher this role, biophysical studies with biomimetic membrane models are often helpful to obtain insights at the molecular and atomic levels. In this review, the added value of biophysics to study lipid-driven biological processes is illustrated using the case of surfactins, a class of natural lipopeptides produced by *Bacillus* sp. showing a broad range of biological activities. The mechanism of interaction of surfactins with biomimetic models showed to be dependent on the surfactins-to-lipid ratio with action as membrane disturber without membrane lysis at low and intermediate ratios and a membrane permeabilizing effect at higher ratios. These two mechanisms are relevant to explain surfactins’ biological activities occurring without membrane lysis, such as their antiviral and plant immunity-eliciting activities, and the one involving cell lysis, such as their antibacterial and hemolytic activities. In both biological and biophysical studies, influence of surfactin structure and membrane lipids on the mechanisms was observed with a similar trend. Hence, biomimetic models represent interesting tools to elucidate the biological mechanisms targeting membrane lipids and can contribute to the development of new molecules for pharmaceutical or agronomic applications.

## 1. Introduction

Knowledge about the interaction of bioactive amphiphilic compounds with plasma membranes is essential for understanding their biological activities. Although these interactions often imply the presence of membrane receptors, some bioactive amphiphilic molecules directly interact with the lipid matrix of membranes [1,2,3]. Among these, two main classes of molecules, rhamnolipids and lipopeptides, have been the subjects of several studies reporting the crucial role of the biomolecule–lipid phase interactions for their antimicrobial activities (for examples see: [2,4,5,6,7,8,9]). While the mechanism is well described for some biological activities of these membrane-active molecules [10,11], for other bioactivities such as plant immunity stimulation by rhamnolipids [12] or lipopeptides [13,14,15], or their antifungal activities [6,16], the mechanism is still elusive.

Among lipopeptides, surfactins, a class of natural lipopeptides produced by *Bacillus* sp., present a broad range of biological properties such as antibacterial [17,18], antitumoral [19,20], antiviral [4,21,22] and antimycoplasmic activities [23] but no antifungal activity [16]. In addition to these activities, surfactins also show ability to prime plants for stronger defense against pathogens by triggering induced systemic resistance, which makes them a promising alternative to chemical compounds for plant protection in sustainable agriculture [24,25,26,27].

The surfactins family consists of pumilacidins, lichenysins and canonical surfactins (SRFs). They are amphiphilic molecules formed by a ring-shaped heptapeptidic backbone closed by a β-hydroxy fatty acid chain. The canonical surfactin is the first described lipopeptide of this family in the literature. In nature, SRF is produced as a mix of homologues with a fatty acid chain variating from 12 to 16 carbons and peptide moiety consisting of L-Glu1-L-Leu2-D-Leu3-L-Val4-L-Asp5-D-Leu6-L-Leu7 [28] (Figure 1). However, variations in the amino acid composition and in the lipid chain structure (isomery and length) gave rise to a large diversity of structures.

It is known that the amphiphilic nature of SRFs favors their interaction with lipid bilayers constituting the cellular membranes, highly suggesting that biological activities of SRFs are mainly related to their interaction with the lipid fraction of the plasma membranes [4,13,14,27]. In the particular case of its plant defense-boosting activity, there is evidence suggesting that SRF sensing at the plant plasma membrane level rely on SRF’s interaction with the lipid fraction of the membrane rather than on membrane protein receptors [13,27]. In support, SRF-eliciting activity is conserved in protease-treated tobacco cells, indicating a lack of involvement of protein receptors, and upon successive SRF treatments, implying that there is no saturation state of high affinity receptors [13,27].

SRFs show potential applications in medicine, agriculture or even the food industry [23,29]. However, their activity is highly impacted by their structure and the lipids constituting membranes [14,30,31]. Thus, an in-depth understanding of their mechanism of action is required to better target the organisms impacted by SRFs and possibly select the most suitable SRF variant (length of acyl chain, peptide structure) for one given application. Although some information about interactions between bioactive molecules and cellular membranes can be obtained using living cells (e.g., membranes dynamics and ordering using fluorescence spectroscopy [32]), artificially-made lipid membranes or biomimetic membranes, are required to obtain precise data at molecular and/or atomic levels (e.g., data on thermodynamical parameters, compound penetration and location into membrane, specific interaction with particular lipid or domain) [33]. Biomimetic membranes have easily adaptable lipid composition which allows specific features such as surface charges, domain lateral organization and transversal asymmetry to be highlighted. The control of lipid composition with biomimetic membranes is the key feature which grants them advantages over natural cellular membranes as study models. Three artificial models are commonly used: liposomes, also known as lipid vesicles, supported bilayers and lipid monolayers [33,34,35]. The most commonly used lipids in these models are phospholipids but some studies with increased lipid composition complexity including sterols and/or sphingolipids can also be found. These models are complementary and each of them possesses its own associated techniques [1,36].

In this review, the information given by biophysics using biomimetic membranes to describe SRF–membrane lipid interactions is thoroughly analyzed. We will mainly focus on how membrane models contributed to decipher the (i) SRF insertion and location into membrane, (ii) the effect of such SRF insertion on membrane properties and integrity and (iii) the influence of biomimetic model lipid composition and SRF structure on these interactions. Finally, the link between biophysical results and SRF biological activities is critically addressed to picture the biological relevance of biomimetic models.

## 2. Global Mechanism of Surfactins–Lipid Membrane Interaction

### 2.1. Spontaneous Insertion of Surfactins at the Interface of the Lipid Membrane

In water and in presence of biomimetic membranes, amphiphile molecules such as SRFs have two competing options: self-aggregate to form micelles or insert from the external medium into the membrane, which is the first and key step in the interaction process. Based on the product of the surfactin membrane–water partition coefficient (K) and its critical micellar concentration (CMC) that is lower than 1 (K × CMC = 0.2), Heerklotz and Seelig [37] predicted that SRFs prefer to form micelles rather than to insert into a lipid lamellar structure such as POPC (1-palmitoyl-2-oleoyl-glycero-3-phosphocholine) vesicles. Nevertheless, SRFs have an affinity for lipid bilayers, i.e., are able to insert into lipid bilayers in a thermodynamically spontaneous and endothermic way and a mechanism predominantly driven by hydrophobic interaction [37,38].

The transversal localization of SRFs into the lipid bilayer is another crucial information related to their mode of action. Several studies based on computational simulations [39], ^2^H-NMR (deuterium solid-state nuclear magnetic resonance) [40] and NR (neutron reflectometry) [41] agree that SRF peptide cycle is located close to the phospholipid headgroups at the lipid–water interface and its acyl chain is deeply inserted in the bilayer hydrophobic core. However, SRF location in the outer or inner sheet and its potential trans-bilayers movement have been poorly studied in the literature [39,41,42]. While NR experiments [41] on supported lipid bilayers showed a total insertion of SRF in the outer sheet without possibility of translocation, computational simulations [39] suggested the ability of SRF molecules to flip–flop through the bilayer. Trans-bilayer motions of SRF were also evidenced at high SRF concentrations but not at low ones by zeta potential measurements on POPC vesicles [42]. The apparent controversial results highlight the importance to keep a critical view on their interpretation. In the work of Shen et al. [41], the partial negative charge of the silica support on which the bilayer model is deposited can limit the passage of SRF in the leaflet in contact with the support due to electrostatic repulsions with the two negative charges of SRF peptide cycle. It is worth noting that the influence of other membrane lipid families such as sterols and sphingolipids on the SRF transversal motion has not yet been considered, even though their presence hugely impacts membrane structural properties.

### 2.2. Surfactins Progressively Disorganize Lipids Constituting Membranes with Increasing Surfactins-to-Lipid Ratio

The capacity of SRFs to destabilize membranes is often mentioned to explain their biological activities, mainly their antimicrobial properties. Several biophysical techniques are used to study membrane destabilization induced by SRFs such as e.g., calcein/carboxyfluorescein leakage experiments [30,31,43,44,45], electron microscopy [46,47], atomic force microscopy (AFM) [48,49,50] or NMR [40,51,52]. Globally, it can be said that the destabilizing effect of SRFs is progressive with the SRF-to-lipid ratio (R_b_) related to the SRF concentration (Figure 2a).

At low R_b_ (e.g., R_b_~0.05–0.1 in the case of POPC liposomes), the insertion of SRF induces curvature stress in the outer leaflet and increases its surface area without increasing the one of the inner leaflet [52]. This positive curvature is observed as corrugations in DPPC (1,2-dipalmitoyl-sn-glycero-3-phosphocholine) bilayers by AFM [50] and is responsible for the increase in the transition temperature from the lamellar liquid-crystalline phase to the inverted-hexagonal phase observed by differential scanning calorimetry (DSC) in DEPE (1,2-dielaidoyl-sn-glycero-3-phosphoethanolamine) bilayer with increasing SRF concentrations [4]. This positive curvature induces a transient membrane failure, due to the flip–flop of some SRF molecules from the outer leaflet to the inner one in order to balance the area of both leaflets [48,52].

As the value of R_b_ increases, membrane destabilization by SRF results in a complete release of the liposome internal compartment without formation of micelles [52]. It is explained by the formation of SRF-rich clusters that can form pores, preventing the membrane to anneal and seal off the leakage [30,53,54,55,56,57]. These clusters are confirmed by AFM imaging on lipid monolayers mixing C15-SRF with DMPC (1,2-dimyristoyl-sn-glycero-3-phosphocholine, 14:0 PC), DPPC (16:0 PC) or DSPC (1,2-distearoyl-sn-glycero-3-phosphocholine, 18:0 PC) [56,57] and on mixed bilayers of DPPC and SRFs (mixture of C13-C14-C15 homologs) displaying ripple structures [50]. This separation of phase matches with the strong interaction energies between SRF molecules compared to their association with lipids, possibly originating from hydrophobic mismatch and favoring the immiscibility of the two components [56]. Such immiscibility of SRF and lipids is corroborated with the appearance of a second phase transition peak in the DSC profile of lipid vesicles with increasing SRF concentrations [47,55] and the presence of two Bragg peaks on SANS (small angle neutron scattering) profile in an SRF and DPPC bilayer mixture [58]. Additionally, voltammetry and conductivity measurements showed the formation of pores in DMPC or DOPG (1,2-dioleoyl-sn-glycero-3-phosphoglycerol) lipid membrane in presence of SRFs [30,54]. Besides the formation of SRF-rich clusters and pores in the membrane, intermediate concentrations of SRFs lead to the apparition of sheet-like membrane fragments [46,47,48], which are thought to be membrane fragments of vesicles [47].

At high $R_{b}$, a threshold value $R_{b}^{sat}$ exists at which the membrane lysis or solubilization into micellar structures is initiated [52]. For SRFs interaction with POPC vesicles, a theoretical $R_{b}^{sat}$ value of 0.2 was determined [37] which was experimentally confirmed by different techniques [40,43,44,52]. In calcein leakage assays, a sharp increase in the amount of released calcein is observed close to this ratio [43,44]. Likewise, in solid-state NMR spectroscopy, isotropic contributions characteristic of a small and mobile object such as micelles [59] appear on ^31^P- and ^2^H-NMR spectra for $R_{b}$ above 0.2–0.3 [40,52]. The beginning of solubilization was also noticed in isothermal titration calorimetry (ITC) profile by a change of the heat peak sign from endothermic to exothermic at $R_{b}$ of 0.22 [52]. The solubilization process is achieved when $R_{b}$ reaches the value of $R_{b}^{sol}$, determined by ITC when the heat of titration returns to values close to zero [52].

## 3. The Interaction of Surfactins with Lipid Membranes Is Influenced by the Lipid Composition Complexity

Single phospholipid models were useful to reveal that SRFs (i) interact spontaneously with membranes, (ii) are located close to the hydrophobic tail/polar heads interface and (iii) progressively disorganize/destabilize membranes with an SRF-to-lipid ratio-dependent process. However, due to the high complexity of native plasma membranes, the influence of lipid composition used in the models must be considered. Indeed, several biophysical parameters as the membrane curvature, the surface charge and the lipid lateral organization are directly affected by the lipid composition. This is why the structure of the hydrophobic core (lipid acyl chain length, lipid phase) and the lipid polar heads are critical parameters that influence the interaction with SRFs (Figure 2a).

### 3.1. Influence of the Hydrophobic Core Structure

The interaction between SRFs and lipid membranes is an entropy-driven process, meaning that this process is mainly driven by hydrophobic interactions [37]. Hence, the composition (presence and proportion of specific lipids such as sterols and sphingolipids) and the structure of the hydrophobic core (length and unsaturation of the lipid chains) of lipid membranes are critical parameters that can influence the effect of SRFs on membranes.

Regarding the influence of lipid chains, comparison of $R_{b}^{sat}$ values on lipid vesicles with different compositions suggests a variation of $R_{b}^{sat}$ depending on the acyl chain length and unsaturation. Indeed, while a value of 0.22 is obtained for POPC (16:0–18:1 PC) vesicles [52], partial solubilization of soy PC (soy phosphatidylcholine) vesicles (mainly composed of 16:0–18:2 PC) by SRFs is observed at $R_{b}$ = 0.15 [46] while sheet-like membrane fragments are already observed at $R_{b}$ = 0.04 with DMPC (14:0–14:0 PC) vesicles [47]. The stronger impact of SRF on membranes with shorter acyl chain length was also evidenced by using DSC, with a more pronounced decrease in the melting temperature in DMPC liposomes than in the DPPC (16:0–16:0 PC) and DSPC (18:0–18:0) liposomes [55]. In addition, studies on SRF insertion in DSPC, DPPC and DMPC monolayers also revealed an increased penetration and a higher miscibility of C15-SRF or SRF homolog mix with the shorter lipids [56,57,60]. Hence, the increase in the lipid acyl chain length is likely to increase the hydrophobic mismatch between SRF and membrane lipids, resulting in a more difficult insertion of SRF and a stronger phase separation between SRF and lipids, as observed in monolayer experiments [56]. Nevertheless, the effect of acyl chain length might also be related to its effect on lipid physical state. Indeed, while a phase separation between C15-SRF and lipids is observed in DMPC monolayer in gel phase, i.e., DMPC in a more ordered organization with a low lateral lipid mobility [57], C15-SRF and lipids become completely miscible when the DMPC monolayer is in fluid phase [56]. Therefore, the more ordered phase favored by longer lipid chain length seems also to hinder the insertion of SRFs.

The impact of lipid phases on SRF effect has also been studied on supported bilayers and on liposomes [13,48,49]. In accordance with the hypothesis of more difficult insertion of SRFs in more ordered phase, it was observed that lipids in fluid phase (Ld) were more sensitive to SRF activity than lipids in gel phase (So) [48,49]. Nevertheless, the effect of lipid phase is less important than the existence of membrane lateral heterogeneities/defects that favors SRF membrane activity. Such hypothesis is supported by the preferential insertion of SRFs at the boundaries between gel and fluid phases observed in DPPC/DOPC (1,2-dioleoyl-sn-glycero-3-phosphocholine) bilayers [48] and the higher affinity of SRFs for vesicles with higher coexistence of phases (affinity for Ld + Lo + So > Lo + So ≈ Ld + So > Ld ≈ Lo) [13].

Besides the lipid chain length, the presence of sterols also affects the properties of membrane hydrophobic core. On one hand, it was observed that the presence of 33–36% mol cholesterol reduces liposome leakage compared to liposomes without sterols [31,45] (difference of leakage between POPC and POPC/chol observed at R_b_ > 0.2). On the other hand, Fiedler and Heerklotz showed that the presence of 10–30% mol of cholesterol or ergosterol did not affect liposome leakage [43] (no difference of leakage between POPC and POPC/chol at R_b_ up to 0.25). These apparent contradictory results at similar R_b_ can be explained by the difference in the incubation time of SRFs (less than 10 min in [43,45] and 1 h in [41]). Indeed, Heerklotz’s group observed a specific behavior with two steps i.e., a fast but limited graded leakage following by a slow process including the formation of pores [43,52,61]. Thus, cholesterol would limit the initial step of leakage, as observed by Carrillo et al. [45], and Oftedal et al. [31], but would not affect the second step of the process, which will become dominant over longer incubation time [43]. The presence of sterols thus modifies the mechanism of leakage, from a global destabilization of the membrane towards a more local action leading to a more specific defect on the structure [43].

### 3.2. Influence of the Membrane Surface

In addition to the membrane hydrophobic core structure, SRF–membrane interaction can also be affected by the polar headgroup nature of the phospholipids used for the model. Indeed, a variation of the lipid polar head nature (with one negative charge for PG (phosphatidylglycerol), PS (phosphatidylserine) and PA (phosphatidic acid) lipids and neutral for PC (phosphatidylcholine) and PE (phosphatidylethanolamine) lipids) affects the surface of the lipid systems by influencing its curvature and its global charge.

The presence of lipids promoting membrane negative curvature such as PE, PA or cardiolipin (CL) induces a reduction of SRF-induced leakage [30,43,45,62]. Due to their inverted conical shape, SRFs introduce a positive curvature stress on the lipid bilayer promoting its disorganization. In consequence, conical lipids such as PE, PA and CL offset the decrease in membrane curvature induced by SRFs and counteracts drastically their destabilization properties. Such counterbalancing between SRF positive curvature and lipid negative curvature also modifies the structure of the lipid membrane as it prevents the emergence of negatively curved hexagonal phase of DEPE membranes [4,55] and stabilizes segregation between SRFs and DPPE (1,2-dipalmitoyl-sn-glycero-3-phosphoethanolamine) in DPPE monolayer [56].

The presence of negatively charged lipids is another parameter affecting the SRF membrane effect. A decrease in SRF penetration is observed in DPPS (1,2-dipalmitoyl-sn-glycero-3-phospho-L-serine) or DPPA (1,2-dipalmitoyl-sn-glycero-3-phosphate) monolayer compared to DPPC monolayer [57,60]. A lower miscibility of C15-SRF in DPPS monolayer than in DPPC monolayer leads to a stronger phase separation between C15-SRF and lipids [56] and a stronger membrane destabilization effect of SRFs is obtained in vesicles composed of DMPS (1,2-dimyristoyl-sn-glycero-3-phospho-L-serine)/DMPC or DMPG (1,2-dimyristoyl-sn-glycero-3-phosphoglycerol)/DMPC compared to pure DMPC vesicles [43,51,55]. The influence of membrane charges is further supported by the lower effect of SRFs on vesicles with DMPS at pH 4.5, when DMPS is in its zwitterionic state, compared to pH 7.5 when DMPS is negatively charged [51]. In this last work, a conversion of DMPC/DMPS multilamellar lipid vesicles (with a diameter size >1µm) into small unilamellar vesicles (with a diameter size of ~70 nm) is observed in the presence of SRFs at pH 7.5 but not at pH 4.5. Hence, after its insertion into the lipid bilayer due to hydrophobic interactions, SRF induces stronger curvature strains, i.e., a higher destabilization effect, in presence of negatively charged lipids due to electrostatic repulsion between the negative charges of SRF peptide moiety and negative lipid headgroups [51].

Finally, the combining effect of membrane negative curvature and negative charges further inhibits the leakage induced by SRFs [43].

## 4. Surfactin’s Structure Influences Their Interactions with Biomimetic Membranes

Alongside the influence of the biomimetic lipid system composition, some studies have used synthetic variants of SRFs to investigate the influence of the SRF structure on their interaction with membranes [13,38,49,63]. Two parameters are crucial for SRF membrane affinity as well as for their membrane destabilization properties: (i) the structure of the SRF peptide moiety (cyclic vs. linear conformation and number of charges) and (ii) the SRF hydrophobic acyl chain length (Figure 2b).

Linearization of the peptide cycle induces a decrease in SRF binding affinity for POPC vesicles [38] and a lower penetration in the DPPC monolayer [57]. It also increases their CMC [63] and reduces their membrane solubilization properties, especially regarding lipids in the gel phase [49]. The stronger activity of the cyclic analog was suggested to be due to its “horse saddle-like structure” which favors the distinction of polar and non-polar regions within the molecule, i.e., reinforces the amphiphilic nature of SRFs [38,63].

In addition to SRF conformation, the number of negative charges in the peptide part also influences SRF membrane activity. Increasing the number of negative charges from two to three in linear SRF analogs induced an important decrease in SRF membrane affinity for POPC vesicles [38]. This result contrasts with the slightly stronger membrane activity observed for the SRF variant with three negative charges, especially marked by its ability to destabilize rigid DPPC domains which is not the case for the variant with two negative charges [49]. To corroborate the role of SRF’s charge on its membrane activity, the naturally occurring SRF-like lipopeptide, lichenysin, is an interesting candidate since it displays a change in the amino acid sequence (replacement of glutamate by glutamine) leading to the presence of 1 negative charge instead of 2. This variant showed a similar CMC as canonical SRF (14.7 µM for lichenysin vs. 2.95 to 12.85 µM for SRFs) [48,52,64]. The quantification of lichenysin binding to lipid membranes and its solubilization properties have not been investigated yet, which complicates a close comparison with canonical SRFs. However, its insertion into the lipid membrane showed similar features as SRFs in terms of concentration- and lipid composition-dependent mechanism [64,65].

Changes in SRF membrane activity in the presence of cations in the aqueous medium such as calcium or sodium also reveal the importance of the SRF peptide charge. Neutralization of the SRF negative charges by cations leads to a stronger SRF penetration in lipid monolayers [57,60], stronger membrane disturbance [55] and a decrease in CMC [66]. The impact is more marked for bivalent cations than monovalent cations [66,67].

Concerning the acyl chain length, it was shown that increasing the chain length of linear analogs decreases the CMC [63] and increases both SRF affinity for membranes [13,38] and their membrane solubilization properties [49]. Thus, increasing the number of carbons in the acyl chain seems to counterbalance the loss of the cyclic feature of the peptide part.

## 5. Relationships between Surfactin Biophysical Properties and Biological Activities

The biophysical studies exposed previously gave valuable insights on the mechanism by which SRFs interact with lipid membranes. At low and moderate concentrations, SRFs tend to disturb membrane structure through curvature stress and modification of membrane lateral organization but without impacting membrane integrity. Then, at higher concentrations, SRFs permeabilize membrane through a detergent effect and/or the formation of pores. The sensitivity of lipid membranes to SRFs is bidirectionally influenced by their lipid composition and SRF’s structure. These parameters govern the penetration ability and the affinity of SRFs for membranes and, consequently, the concentration threshold ($R_{b}^{sat}$) at which SRFs destabilize lipid membranes. As with biomimetic models, the interaction of SRFs with native biological membranes also showed either a cell lysis effect by a detergent action and/or a pore formation [31,63,68] or a disturbance of membrane structure without lysis induction [4,13,14,69,70].

Membrane permeabilization is the main and most common mechanism to explain the antibacterial and hemolytic activities of SRFs [2,29,37,71] (Figure 3a). In the same way as what has been observed in biomimetic models, both antibacterial and hemolytic activities are affected by SRF structure [21,63,72] and the lipid composition of the native biological membranes [30,31,62,73]. The linearization of SRF peptide moiety or a decrease in SRF acyl chain length lead to a reduced hemolytic activity [21,63,72]. The neutralization of the peptide charges also decreases the hemolytic activity but the reduction from 2 to 1 negative charge slightly increases it [21]. Beside, a stronger SRF effect was observed on cholesterol-depleted red blood cells [31] and on *Bacillus subtilis* strain with lower cardiolipin content [30]. In the latter case, it was observed that in presence of SRFs, *B. subtilis* tends to have a higher content in lipids promoting negative curvature (cardiolipin and PE), where this change is SRF concentration dependent [62,73]. Supported by the lower SRF-induced leakage observed on liposomes containing cardiolipin or PE, these results shed light on the importance of negative curvature lipids in bacterial tolerance to SRFs [30,62,73].

The action of SRFs as membrane disturbers was suggested to explain the activity of SRFs on tumorous cells at concentrations where no lysis is observed [69,70]. The absence of viral membrane disruption or plant cell lysis at SRF concentrations showing antiviral [4] or plant immunity-eliciting activity [13,74] has also been noticed. Therefore, these last activities seem to also be more related to the action of SRFs as membrane structure disturbers (Figure 3b) rather than to their permeabilizing properties. As for the permeabilization mechanism, the effect of SRFs on membrane structure is dependent on SRF structure and lipid composition of the native biological membrane. The loss of the cyclic structure in peptide moiety, the neutralization of the peptide charges and a shorter SRF chain length lead to a lower effect on plant cells and viruses [13,21] while the presence of one negative charge instead of two leads to a slightly increased antiviral activity [21]. Biophysical data on biomimetic membranes combined with biological results obtained on plant cells and viruses showed that the immunity-eliciting and antiviral effects are achieved by SRFs without membrane disruption, emphasizing the general importance of SRF–lipid interactions beyond their most known mechanism, membrane disruption. In the case of SRF’s antiviral activity, membrane models illustrated that SRFs counterbalance the negative curvature in the viral envelope. This mechanism is likely to hinder the membrane fusion required for viral genome release into the host cells [4]. In the case of the plant-eliciting activity, membrane models highlighted the tendency of SRFs to interact with the biomimetic membrane lipid phase, thus giving hints for deciphering SRF’s interaction with biological membranes and their sensing by plant cells [13,14,74]. Nevertheless, the exact molecular mechanism responsible for the triggering of the defense signaling by the SRFs through the membrane is not yet understood.

In both mechanisms, the membrane permeabilization and membrane structure disturber, biomimetic models proved to be relevant regarding biology and contributed to decipher specific features of the mechanism difficult to study on complex native biological membranes. However, strict parallelism between the biophysical and biological studies is sometimes complicated. One reason is that the SRF-to-lipid ratio (R_b_), a key parameter in biophysical studies to distinguish the different steps in the process of the SRF–membrane interaction, is very difficult to be determined in biological studies. Nevertheless, it can be noticed that the biological activities of SRFs related to the membrane structure disturbance occur at lower concentrations (around 2.5–10 µM for plant immune stimulation [74] and around 10 µM for antiviral activity [4]) than the one related to its detergent effect (around 20–30 µM for hemolytic, antibacterial and antimycoplasmic activity [4,5,63,75]). Hence the gradual interaction process separated in two main phases (i.e., membrane structure disturbance and membrane solubilization) observed with liposomes is also biologically relevant.

Additionally, membrane models also contributed to a better understanding of the structure–function relationships of SRF biological activities. SRFs with a shorter chain length have reduced affinity for membrane models and lower membrane solubilization properties, likely due to a decrease in hydrophobic interactions between SRFs and the lipid bilayer. Such decrease in activity for SRFs with shorter chain length was also observed on viruses, plants and red blood cells [13,21,63,72]. The linearization of the SRF peptide cycle or the neutralization of SRF amino acid negative charges decreases their antiviral, hemolytic and plant-eliciting activities [13,21,63] while the presence of a single negative charge increases their antiviral activity [21], similarly to the trend observed for their membrane activity on membrane models.

## 6. Biomimetic Models to Solve Challenges in Biology

In this paper, we highlighted the potential of biomimetic membrane models to study biological mechanisms of membrane-active molecules such as SRFs and decipher their key structural features.

However, as a simplified version of native biological membranes, membrane models do not allow the study of an entire biological mechanism of perception or interaction of a bioactive molecule. Although, when used to study more specific features of mechanism related to SRF–lipid interaction, such as the effect of lipid composition or the importance of SRF structural traits, results obtained on these models are strongly correlated with biological facts, illustrating their biological relevance. Biomimetic membrane models are thus very interesting tools for researchers to investigate membrane lipid-dependent activities at atomic and molecular scales. Nevertheless, the strong influence of lipid composition on the results illustrates that the lipid composition of the model must be representative of the organism studied. Sufficient lipidomic knowledge is thus required to establish an adapted biomimetic model. However, a balance between simplicity and complexity of the models (higher number of lipids, asymmetry between the outer and inner leaflets) is always pursued in biophysics to obtain unambiguous interpretable results and valuable information on biological activities.

The information given by biophysics on biomimetic models is also very useful for targeting the best application for development purposes in the pharmaceutical or agronomic industries [76,77]. Additionally, the integrative biophysical analysis of membrane-interacting molecules using biomimetic models could also be useful to study and develop other membrane-active compounds, such as other lipopeptides, rhamnolipids or virus fusion inhibitors [2,78,79], and will hence help to overcome current and future challenges in biology.

## Figures and Tables

**Figure 1 ijms-23-13831-f001:**
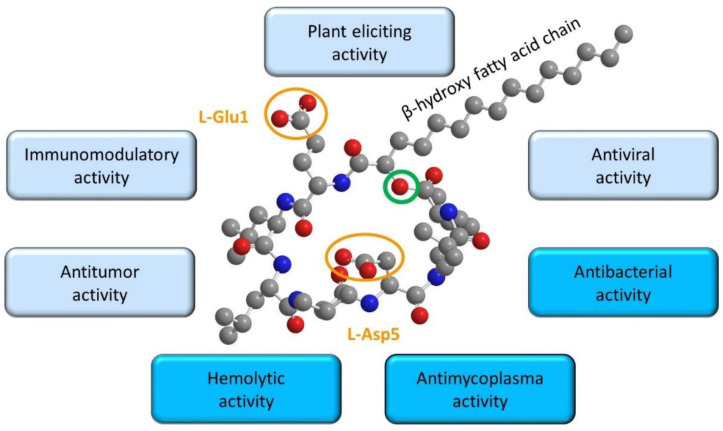
Structure and main biological activities of canonical surfactin (SRF). The SRF structure is composed of a fatty acid chain variating from 12 to 16 carbons (C15 represented here) and a peptide moiety consisting of L-Glu1-L-Leu2-D-Leu3-L-Val4-L-Asp5-D-Leu6-L-Leu7 represented in 3D (using the software Chemdraw professional and Chem3D) with the charged groups encircled in orange and the lactone bond encircled in green. The activities associated with the well-known membrane permeabilizing effect are framed in dark blue, the activities for which the mechanism is still under investigation are in light blue.

**Figure 2 ijms-23-13831-f002:**
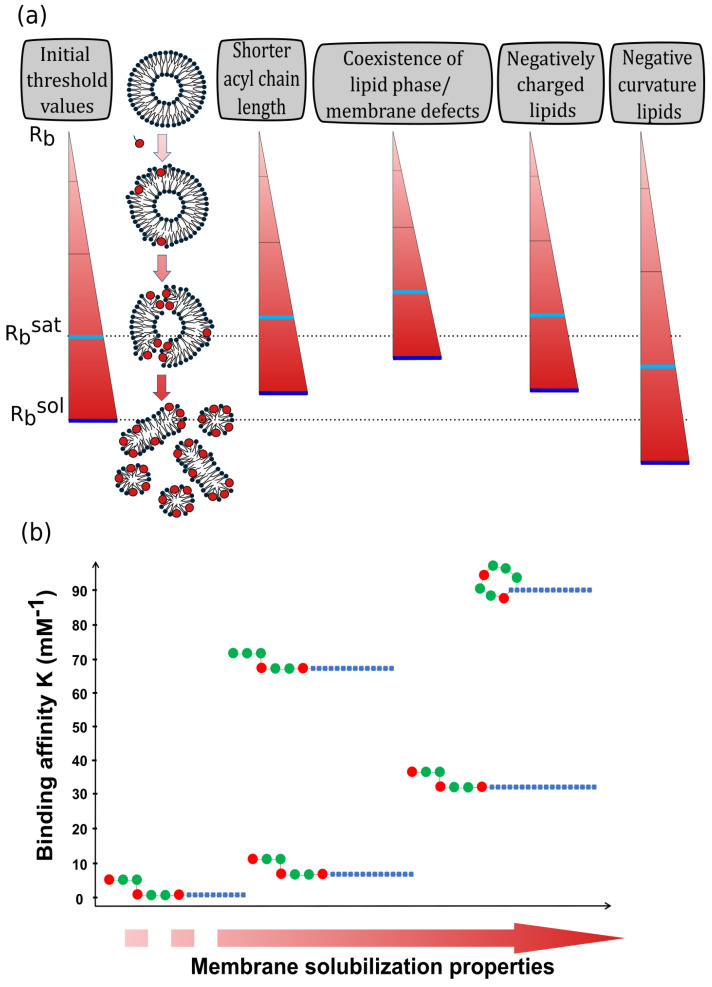
The progressive destabilizing effect of surfactin on lipid membrane and the parameters influencing it. (**a**) Effect of membrane lipid composition and structure on the threshold value of the surfactin-to-lipid ratio (R_b_) leading to the beginning of lipid membrane disruption (R_b_^sat^, light blue line) and its total solubilization (R_b_^sol^, dark blue line). (**b**) Effect of surfactin structure on its affinity for lipid membranes and its membrane solubilization properties. Symbols for surfactin variants: blue squares are the carbon atoms of the acyl chain, red disks are charged amino acids and green disks are neutral amino acids.

**Figure 3 ijms-23-13831-f003:**
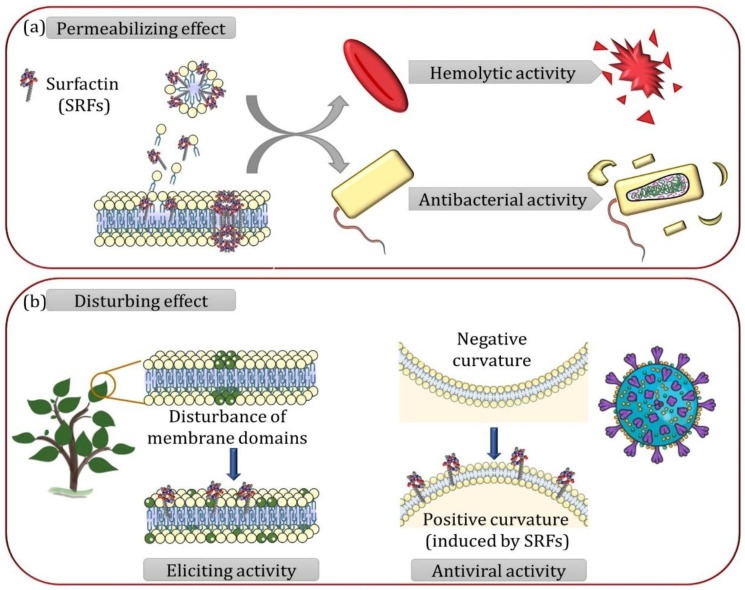
Schematic representation of the molecular mechanism related to the biological activities of surfactins (SRFs) (**a**) SRFs permeabilizing effect. Detergent effect and pores formation are associated with the hemolytic and antibacterial activities of SRFs. (**b**) SRFs membrane disturbing effect. In plants, disturbance of membrane domain organization is associated to the eliciting activity of SRFs. The SRF antiviral activity is associated with its capacity to induce a positive curvature counterbalancing the negative curvature of the viral envelope required for the membrane fusion between the virus and the host cell.

## Data Availability

Not applicable.

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
