# Peer review of "Added Value of Biophysics to Study Lipid-Driven Biological Processes: The Case of Surfactins, a Class of Natural Amphiphile Molecules"

_ijms, 2022, doi:10.3390/ijms232213831_

Round 1
Reviewer 1 Report
Guillaume Gilliard and colleagues present a well written, easy to follow and very comprehensive review that encompasses a detailed description of surfactins’ effects on both biomimetic membrane models and biological membranes. A considerable amount of evidence obtained using different biophysical approaches is given by the Authors to show the main molecular mechanisms by which surfactins attained their effects. Moreover, the Authors do not simply make a report of the literature; occasionally, they offer their critical view over the published works. There are only few comments that must be addressed prior to publication.
Comments
11) Add Rbsat and Rbsol to the list of abbreviations.
22) When presenting and discussing the effects of surfactins in lipid bilayers, especially when considering the hydrophobic mismatch (e.g., lines 214 – 216) between surfactin and membrane lipids it might be interesting to mention the chain length of the surfactin if a mixture was not used, which is the case of the work in reference 56.
33) Paragraph lines 233 – 246: It is important to mention the surfactin-to-lipid ratio used in the works cited throughout this paragraph.
44) Paragraph in lines 254 – 263 is a bit confusing and unclear:
a. “decrease of SRF penetration in DPPS or DPPA monolayer”; “a lower miscibility of SRF in DPPS monolayer leading to a stronger phase separation between SRF and lipids”; “a stronger membrane destabilization in vesicles with DMPS or DMPG” – In all these descriptions where it is stated that there is a “decrease of…” or that “a lower…” or “a stronger…” the reader does know as compared to what?
b. “The influence of membrane charges is further supported by the lower effect of SRFs on vesicles with DMPS at pH 4.5, when DMPS is in its zwitterionic state, compared to pH 7.5 when DMPS is negatively charged [51]. Hence, the electrostatic repulsion plays an important role in SRF activity on negatively charged membranes.” – The Authors need to explain a little better the outcome of this work, since in its current form is confusing because it is mentioned that there is a lower effect at pH 4.5 (when the global charge of the lipid is neutral) than at pH 7.5 (when the lipid is negatively charged) and at the same time it is said that electrostatic repulsion plays a role in the activity of SRF on negatively charged membranes. When one reads this last part, it would expect that due to electrostatic repulsion the effect on negatively charged membranes would be lower than on globally neutral membranes. This is why I ask the Authors to explore a little bit more the work of reference 51 by detailing some of the main results.
55) In figure 3b the part of the negative curvature is not a zoom in of the viral envelope in the conditions shown in the figure. Membrane negative curvature is observed during membrane fusion which is not what the Authors are showing. I would ask the Authors to either replace the virus schematic representation for a membrane fusion scheme or just erase the brownish lines and circle that give the idea of a zoom in.
66) Lines 381 – 382: “occur at lower concentrations (around 2.5-10 μM for plant immune stimulation[74] and around 10 μM for antiviral activity[4] than the one related to its detergent effect (around” – The first parenthesis is not closed after “…activity[4]”.
Author Response
Dear Editor and reviewers,
On behalf of my co-authors and myself, I would like to thank you and the reviewers for the comments. We sincerely appreciate the thorough analysis of the work as well as the timely fashion with which the report has been communicated to us.
We have analysed all the criticisms raised about the work and modified the manuscript to take into account all the suggestions.
The changes are marked up using the “Track Changes” function.
Hereafter, you will find our responses to both the reviewer and the editor’s comments, raised point by point.
We hope that with these changes made, our contribution will be acceptable for publication in IJMS.
Best regards,
Magali Deleu
1) The list of abbreviation was completed
2) When it was relevant, we mentioned the chain length of SRF : see lines 161, 163, 215, 221, 222, 284 and 285 in the « track changes » version.
3) The surfactin-to-lipid ratios were added in the text : see lines 239 and 241-242 in the « track changes » version.
4.a) The comparison systems were added in the text : see lines 284, 285, and 288-289 in the « track changes » version.
4.b) A short description of the work of reference 51 was added : see lines 291-297 in the « track changes » version.
5) We erased the brownish lines and circles in the new version of fig 3.
6) The parenthesis was added : see line 406 in the « track changes » version.
Reviewer 2 Report
This review deals with the “Added value of biophysics to study lipid-driven biological processes: the case of surfactins, a class of natural amphiphile molecules.” This paper covers efficiently the most recent references directly concerning the subject. The text describes critically both the composition and mechanism of action of surfactins using biomimetic models. It also relates the biophysical findings to known biological functions of the lipopeptides. This reviewer considers the text and the contents adequately presented for publication in this journal.
Author Response
Dear Editor and reviewers,
On behalf of my co-authors and myself, I would like to thank you and the reviewers for the comments. We sincerely appreciate the thorough analysis of the work as well as the timely fashion with which the report has been communicated to us.
We have analysed all the criticisms raised about the work and modified the manuscript to take into account all the suggestions.
The changes are marked up using the “Track Changes” function.
Hereafter, you will find our responses to both the reviewer and the editor’s comments, raised point by point.
We hope that with these changes made, our contribution will be acceptable for publication in IJMS.
Best regards,
Magali Deleu
